# Structure Learning with Side Information: Sample Complexity

**Saurabh Sihag**                    **Ali Tajer**

Electrical, Computer, and Systems Engineering Department
Rensselaer Polytechnic Institute

## Abstract

Graphical models encode the stochastic dependencies among random variables (RVs). The vertices represent the RVs, and the edges signify the conditional dependencies among the RVs. Structure learning is the process of inferring the edges by observing realizations of the RVs, and it has applications in a wide range of technological, social, and biological networks. Learning the structure of graphs when the vertices are treated in isolation from inferential information known about them is well-investigated. In a wide range of domains, however, often there exist additional inferred knowledge about the structure, which can serve as valuable side information. For instance, the gene networks that represent different subtypes of the same cancer share similar edges across all subtypes and also have exclusive edges corresponding to each subtype, rendering partially similar graphical models for gene expression in different cancer subtypes. Hence, an inferential decision regarding a gene network can serve as side information for inferring other related gene networks. When such side information is leveraged judiciously, it can translate to significant improvement in structure learning. Leveraging such side information can be abstracted as inferring structures of distinct graphical models that are *partially* similar. This paper focuses on Ising graphical models, and considers the problem of simultaneously learning the structures of two *partially* similar graphs, where any inference about the structure of one graph offers side information for the other graph. The bounded edge subclass of Ising models is considered, and necessary conditions (information-theoretic ), as well as sufficient conditions (algorithmic) for the sample complexity for achieving a bounded probability of error, are established. Furthermore, specific regimes are identified in which the necessary and sufficient conditions coincide, rendering the optimal sample complexity.

## 1  Introduction

Graphical models are widely used to compactly model the conditional interdependence among multiple random variables Lauritzen [1996] and Pearl [2009]. The vertices of the graph represent the random variables (RVs), while the edges encode the inter-dependence among the RVs. The complete structure of the graph is analytically captured by the joint probability distribution of the random variables. Graphical models offer effective and tractable solutions to various inferential and decision-making solutions in different domains, e.g., computer vision Won and Derin [1992], genetics  Chen et al. [2013], Fang et al. [2016], Dobra et al. [2004], social networks  Jacob et al. [2014], and power systems Dvijotham et al. [2017]. In this paper, we focus on Ising models and consider the problem of *joint model selection* of a pair of graphs with partially identical structures using the samples from their joint distributions.

Graphical models with partially similar structures arise in the domains that consist of multiple layered networks of information sources. In such an application, each layer shares some of its vertices

and the data it generates with other layers that contain the same vertices. For example, the gene networks that represent the subtypes of the same cancer share similar edges across all subtypes and also have unique edges exclusive to each subtype of cancer Chen et al. [2013]. In a different context, a similar problem emerges in analyzing the voting patterns of the members of the US Senate Guo et al. [2015] for different categories of bills, where the statistical models reveal the common dependency structure across the members affiliated to the same political party and other structures unique to each class. Similarly, in the context of social networks, the relationships among a group of individuals on different platforms (e.g., Twitter and Facebook) form two distinct, but potentially partially similar graphical models. In such applications, learning one graph provides a significant amount of information that can be used for learning other related graphs.

Due to the costs associated with collecting and processing data samples in large-scale graphical models, it is of interest to study the sample complexity of learning multiple structures simultaneously, where inference about each structure serves as side information for other structures.

## 1.1 Related Work

While the problem of graph structure learning is NP-hard Chickering [1996], it becomes feasible under certain restrictions on the structure of the graph. For instance, the studies in Yuan and Lin [2007], Rothman et al. [2008], Ravikumar et al. [2010], Banerjee et al. [2008] investigate recovering the structure of the graphical model under sparsity. Such conditions on the structures of the graphical models can be analyzed theoretically by considering certain restricted sub-classes of graphical models, for e.g., graphs with a bounded degree or bounded number of edges.

Information-theoretic analysis of structure learning establishes the algorithm-independent difficulty of recovering the structure of different classes of graphs. The studies in Santhanam and Wainwright [2012], Tandon et al. [2014], Scarlett and Cevher [2016] characterize the necessary conditions on the sample complexity of selecting the model of a given graph in various sub-classes of Ising models. In Santhanam and Wainwright [2012], the necessary and sufficient conditions on the sample complexity for the exact recovery of the graph are established for the class of Ising models under restrictions on the maximum degree and the maximum number of edges in the graph. The results in Santhanam and Wainwright [2012] are extended to a set based graphical model selection in Vats and Moura [2011], where the graph estimator outputs a set of potentially true graphs instead of a unique graph. Similarly, necessary conditions on the sample complexity are established for girth-bounded graphs and path restricted graphs in Tandon et al. [2014]. In Scarlett and Cevher [2016], the problem of graphical model selection is studied for various sub-classes of Ising models under the criterion of approximate recovery. In Das et al. [2012], approximate recovery bounds on the sample complexity are characterized for Ising and Gaussian models without considering the effect of edge weights. The information-theoretic bounds on the sample complexity for structure recovery in Gaussian models are established in Wang et al. [2010], and the information-theoretic bounds for structure learning in power-law graph class are characterized in Tandon and Ravikumar [2013].

Algorithm-independent bounds on the sample complexity have also been investigated for other inference tasks besides model selection from the samples. In Gangrade et al. [2017], the problem of detecting whether two Markov network structures are identical or different is studied, and sample complexity is characterized. The problem of property testing for Ising models is investigated in Neykov and Liu [2017], and information-theoretic limits for testing graph properties such as connectivity, cycle presence, and maximum clique size are established. In Devroye et al. [2018], the problem of density estimation using the samples from the Ising model, is investigated, and the minimax rate of estimation is analyzed.

Joint inference of multiple graphical models, even though recognized as an inference problem that arises in various domains, is primarily studied only algorithmically in Chen et al. [2013], Fang et al. [2016], Guo et al. [2011], Danaher et al. [2014], Mohan et al. [2014], Yang et al. [2015], Peterson et al. [2015], Guo et al. [2015], Qiu et al. [2016]. In Chen et al. [2013], an empirical Bayes method, is deployed to identify interactions that are unique to each class and that are shared across all classes. In Fang et al. [2016], Guo et al. [2011], Danaher et al. [2014], Mohan et al. [2014], Yang et al. [2015] graphical Lasso-based algorithms are designed for joint inference of Gaussian graphical models. An optimization framework is used in Guo et al. [2015] for joint estimation of the graph structures based on discrete data. Similarly, Peterson et al. [2015] investigates the problem of joint estimation of

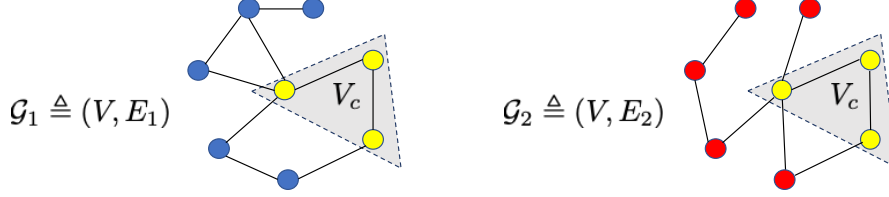

Figure 1: Partially similar structures. Yellow nodes in both graphs have identical structures ($p = 8, q = 3$).

Gaussian graphical models using a Bayesian approach, where the data groups are used to identify partially similar models, and their similarity is leveraged.

## 1.2 Contributions

All the studies above on joint graphical model inference propose empirical or algorithmic model-based frameworks. In this paper, in sharp contrast, we provide an information-theoretic perspective for jointly learning the structures of a pair of similar Ising models. Such analysis offers algorithm-independent *necessary* conditions on the sample complexity for achieving any arbitrary level of reliability in the inference decision. In our previous work in Sihag and Tajer [2019] we considered a sparsely connected, path-restricted sub-class in the context of Ising models and established the algorithm-independent necessary conditions on the sample complexity. In this paper, we consider a more general sub-class of edge bounded Ising models and provide the necessary conditions for all feasible values of the number of edges in the graphs. Furthermore, we also analyze a maximum likelihood (ML)-based graph decoder to establish *sufficient* conditions on the sample complexity.

Based on these bounds, we also provide the asymptotic scaling behavior of these conditions in different regimes. These analyses, as a by-product, also recover the existing relevant results on the recovery of single graphs in Scarlett and Cevher [2016]. Finally, we provide numerical evaluations of ML-based decoder to study the effect of structural similarity on its performance.

A structurally similar pair of graphs are assumed to have identical connectivities in a subgraph formed by a known cluster of nodes. Such settings have been analyzed extensively in the context of seeded graph matching and alignment problems Fishkind et al. [2019], Lyzinski et al. [2014], where in contrast to this paper, the focus is on aligning the vertices of a partially aligned pair of graphs.

## 2 Graph Model

Consider two[1] undirected graphs $\mathcal{G}_1 \triangleq (V, E_1)$ and $\mathcal{G}_2 \triangleq (V, E_2)$, such that the graphs are formed by the same set of vertices $V \triangleq \{1, \ldots, p\}$ but have distinct sets of edges, denoted by $E_1 \subseteq V \times V$ and $E_2 \subseteq V \times V$. When there exists an edge between nodes $u, v \in V$ in graph $\mathcal{G}_i$, we denote it by $(u, v) \in E_i$. Since the graphs are undirected, we have $(u, v) = (v, u)$. We also define the set $\mathcal{N}_i(u) \in V$ as the neighborhood of node $u$ in graph $\mathcal{G}_i$, i.e.,

$$\mathcal{N}_i(u) \triangleq \{w \in V : (u, w) \in E_i\} . \tag{1}$$

It is assumed that a pre-specified cluster of $q$ nodes denoted by $V_c \subseteq V$ have identical internal graph structures in both $\mathcal{G}_1$ and $\mathcal{G}_2$. An example of two such graphical models is illustrated in Fig. 1.

We assume Ising models for both graphs $\mathcal{G}_1$ and $\mathcal{G}_2$, where we define $X_i^u \in \mathcal{X} \triangleq \{-1, 1\}$ as the random variable associated with the node $j \in V$ in graph $\mathcal{G}_i$, for $i \in \{1, 2\}$. Accordingly, one sample from graph $\mathcal{G}_i$ is given by the random vector $\mathbf{X}_i \triangleq [X_i^1, \ldots, X_i^p]$. The joint probability density function (pdf) of $\mathbf{X}_i$ associated with the graph $\mathcal{G}_i$ is given by

$$f_i(\mathbf{X}_i) = \frac{1}{Z_i} \exp \left( \sum_{u,v \in V} \lambda_i^{uv} X_i^u X_i^v \right) , \tag{2}$$

where

$$\lambda_i^{uv} = \begin{cases} \lambda, & \text{if } (u,v) \in E_i \\ 0, & \text{otherwise} \end{cases} , \tag{3}$$

and $Z_i$ is the partition function, given by

$$Z_i = \sum_{\mathbf{X}_i \in \{-1,1\}^p} \exp\left(\sum_{u,v \in V} \lambda_i^{uv} X_i^u X_i^v\right) . \tag{4}$$

Throughout the rest of the paper, we refer to $\mathbf{X}_i$ as one graph sample. The parameter $\lambda \in \mathbb{R}^+$ defined in (3) captures the interdependency among the random variables associated with the vertices. We remark that as $\lambda$ grows or diminishes, i.e., in the asymptote of large or small values of $\lambda$, it becomes increasingly difficult to distinguish the two distinct Ising models Santhanam and Wainwright [2012]. Finally, corresponding to graph $\mathcal{G}_i$, we also define the maximum neighborhood weight according to

$$\zeta_i \triangleq \max_{w \in V} \sum_{u \in \mathcal{N}_i(w)} \lambda_i^{wu} . \tag{5}$$

## 3 Joint Structure Learning with Side Information

In this section, we formalize the notation of similar graphical models with partially identical structures, the recovery criterion, and the associated performance measures.

### 3.1 Graph Similarity Model

**Definition 1.** *Two graphs $\mathcal{G}_1$ and $\mathcal{G}_2$ with identical subgraphs with $q$ nodes are said to be $\eta-$similar, where $\eta = \frac{q}{p}$.*

For given $\mathcal{G}_1$ and $\mathcal{G}_2$, the edges between a pair of nodes with at least one node not in $V_c$ are assumed to be structurally independent of each other. We denote the class of Ising models by $\mathcal{I}$, and the class of $\eta-$similar pairs of Ising models by $\mathcal{I}_\eta$. In this paper, we focus on an *edge-bounded* sub-class of Ising models defined next.

**Definition 2.** *This edge-bounded class of all the $\eta-$similar pair of graphs $\mathcal{G}_1$ and $\mathcal{G}_2$ is specified by parameters $k \in \mathbb{N}$ and $\gamma \in (0,1)$. The maximum number of edges in each graph is $k$ and the number of edges in the identical subgraphs is $\gamma k$.*

Note that in the definitions above the choices of $\gamma$ and $\eta$ are not independent. Clearly, for any combination of $k$ and $p$, $\gamma$ should satisfy $\gamma k \leq \binom{q}{2}$.

For convenience in notations, we also define $\bar{q} \triangleq p - q$ and $\bar{\gamma} \triangleq 1 - \gamma$. It is also assumed that the maximum neighborhood weight, defined in (5), is upper bounded by $\log \zeta$, i.e., $\zeta_i \leq \log \zeta$, for $i \in \{1,2\}$. Finally, we remark that all the results provided for the edge-bounded class have counterparts for the *degree-bounded* class as well, which due to space limitations are omitted.

### 3.2 Recovery Criterion and Figure of Merit

The objective is to jointly estimate the structures of graphs $\mathcal{G}_1$ and $\mathcal{G}_2$ based on a collection of $n$ independent samples generated by each graph. The collection of $n$ graph samples from the graph $\mathcal{G}_i$ is denoted by $\mathbf{X}_i^n \in \mathcal{X}^{n \times p}$. We define the graph decoder

$$\psi : \mathcal{X}^{n \times p} \times \mathcal{X}^{n \times p} \to \mathcal{I}_\eta , \tag{6}$$

as a function that maps the collection of samples to the graphs in class $\mathcal{I}_\eta$. We assume that in each recovered graph we can tolerate erroneous decisions about at most $d$ number of edges, where $d$ is pre-specified. To capture the accuracy of such decisions, we define $\mathsf{P}(\mathcal{I}_\eta, d)$ as the maximal probability of error over the class $\mathcal{I}_\eta$, i.e.,

$$\mathsf{P}(\mathcal{I}_\eta, d) \triangleq \max_{\mathcal{G}_1, \mathcal{G}_2 \in \mathcal{I}_\eta} \mathbb{P}\left[\max_{i \in \{1,2\}} \{|E_i \Delta \hat{E}_i|\} \geq d\right] , \tag{7}$$

where $|E_i \Delta \hat{E}_i|$ is the edit distance between $E_i$ and the estimated edge structure $\hat{E}_i$ given by

$$|E_i \Delta \hat{E}_i| \triangleq |(E_i \backslash \hat{E}_i) \cup (\hat{E}_i \backslash E_i)| . \tag{8}$$

Therefore, $|E_i \Delta \hat{E}_i|$ represents the number of edges to be inserted or deleted to transform $E_i$ to $\hat{E}_i$. Also, $d$ represents the distortion level of the estimated graphs with respect to the true graphs.

## 4  Sample Complexity: Main Results

In this section, we provide the sufficient and necessary conditions on the sample size $n$ for any graph decoder to recover a pair of graphs with bounded probabilities of error. The necessary conditions established are algorithm-independent and characterize the performance benchmarks on the sample complexity for any designed algorithm. The sufficient conditions determine the feasibility of graph recovery under the proposed recover algorithm (ML decoding) under given decision reliability constraints.

A summary of some of the main observations is provided in Table 1.

Table 1: Summary of the main results for recovering Ising models of class $\mathcal{I}_\eta$.

| Parameters | Approx. recovery ($d > 0$) (Necessary conditions) | Approx. recovery ($d > 0$) (Sufficient conditions) | Exact recovery ($d = 0$) (Necessary conditions) |
|---|---|---|---|
| $\lambda = O\left(\frac{1}{\sqrt{k}}\right)$ $k = O(p)$ | $\Omega(k \log p)$ | $\Omega(k^2 \log p)$ | $\Omega(k \log p)$ |
| $\lambda = O\left(\frac{1}{\sqrt{k}}\right)$ $k = \Omega(p)$ and $k = O(p^{\frac{4}{3}})$ | $\Omega(k)$ | $\Omega(k^2 \log p)$ | $\Omega(k \log p)$ |
| $\lambda = O\left(\frac{1}{\sqrt{k}}\right)$ $k = \Omega(p^{\frac{4}{3}})$ and $k = O(p^2)$ | $\Omega(\frac{p^2}{\sqrt{k}})$ | $\Omega(k^2 \log p)$ | $\Omega(k \log p)$ |
| $\lambda = O\left(\frac{1}{p}\right)$ $k$ fixed and $k \leq p/4$ | $\Omega(p^2 \log p)$ | $\Omega(p^2 \log p)$ | $\Omega(p^2 \log p)$ |

### 4.1  Sufficient Conditions

In order to establish sufficient conditions, we adopt the ML graph decoder defined as

$$(\hat{\mathcal{G}}_1, \hat{\mathcal{G}}_2) \triangleq \arg \max_{(\mathcal{G}_1, \mathcal{G}_2) \in \mathcal{I}_\eta} f_{\mathcal{G}_1, \mathcal{G}_2}(\mathbf{X}_1^n, \mathbf{X}_2^n) . \tag{9}$$

The ML decoder is optimal under the exact recovery criterion, i.e., when $d = 0$ Santhanam and Wainwright [2012]. Under approximate recovery, however, no error is declared if the estimates of the two graphs using the ML decoder lie within $d$ distortion level of the true graphs. We use large deviations analysis of the probability of error of (9) under approximate recovery to analyze its performance.

**Theorem 1** (Class $\mathcal{I}_\eta$). *Consider a pair of $\eta-$similar graphs $\mathcal{G}_1$ and $\mathcal{G}_2$ in class $\mathcal{I}_\eta$. If the sample size $n$ satisfies*

$$n \geq r \max\{A_1, 2A_2\} , \tag{10}$$

*where we have defined*

$$r \triangleq \frac{3\zeta^2 + 1}{\sinh^2(\lambda/4)} , \tag{11}$$

$$A_1 \triangleq \left[(2k' + \gamma k) + \log(2k' - d) + 2(k' + 1)\log p + \log \frac{4}{\delta}\right] , \tag{12}$$

$$A_2 \triangleq \left[(2k' + \gamma k) + \log(2\gamma k - d) + 2(\gamma k + 1)\log q + \log \frac{2}{\delta}\right] , \tag{13}$$

$$k' \triangleq \min\left\{k, \frac{\bar{q}(\bar{q} - 1)}{2} + q\bar{q}\right\} , \tag{14}$$

*then there exists a graph decoder $\psi : \mathcal{X}^{n \times p} \times \mathcal{X}^{n \times p} \to \mathcal{I}_\eta$ that achieves $\mathsf{P}(\mathcal{I}_\eta, d) \leq \delta$.*

Note that $k'$ defined in (14) counts the maximum number of edges that can exist in the graphs after excluding those in the shared identical subgraphs.

In order to gain more insight into the sufficient condition in (10), we evaluate the scaling behavior of the sufficient conditions for $n$ in terms of parameters $\lambda$ (parameter of Ising model in (3)), $\zeta$ (controls maximum neighborhood bound), and $k$ (maximum number of edges in each graph). In all these regimes, it is assumed that $k$ is increasing with the graph size $p$. Furthermore, it can be readily verified that $d$, i.e., the number of errors tolerated by the decoder for the structure of each graph, does not affect the asymptotic scaling behavior of the sample complexity.

1. $\lambda = \Theta(1)$: When the size of identical subgraphs dominates the sizes of non-identical parts such that $\frac{\bar{q}}{q} \ll 1$ and $\gamma k \gg \binom{\bar{q}}{2} + \bar{q}q$, the sample complexity is dominated by $2rA_2$, which scales according to $\Omega(\zeta^2 k \log p)$. Also, when we have $k' = k$, the bound on the sample complexity scales according to $\Omega(\zeta^2 k \log p)$. Therefore, in this regime, under fixed $\delta$, the bound on sample complexity is always dominated by a term that has a scaling behavior given by $\Omega(\zeta^2 k \log p)$.

2. $\lambda = O(\sqrt{k^{-1}})$: By noting that $\sinh(\lambda/4) = \Omega(\lambda)$, in this regime, Theorem 1 implies that there exists a constant $c > 0$ such that when $n > c \cdot \zeta^2 k^2 \log p$, there always exists a graph decoder that achieves $\mathsf{P}(\mathcal{I}_\eta, d) \leq \delta$. If $\zeta = O(\exp(\lambda\sqrt{k}))$, then the bound on sample complexity scales as $\Omega(k^2 \log p)$ for fixed $\delta$.

3. $\lambda = \Theta(\sqrt{k})$: In this regime, when both $\lambda$ and $k$ are increasing with $p$, the bound on the sample complexity scales as $\Omega(\zeta^2 \log p)$. When we have $\zeta \geq \exp(\lambda\sqrt{k})$ and $\lambda\sqrt{k} = \omega(\log(\log p))$, the bound on sample complexity scales exponentially according to $\exp(\lambda\sqrt{k})$.

### 4.2 Necessary Conditions

For describing the results in this subsection, we denote the binary entropy function by

$$ h(\theta) \triangleq -\theta \log \theta - (1 - \theta) \log(1 - \theta), \quad \text{for } \theta \in (0, 1) . \tag{15} $$

**Theorem 2** (Class $\mathcal{I}_\eta$ with $k \leq p/4$)**.** *Consider a pair of $\eta-$similar graphs $\mathcal{G}_1$ and $\mathcal{G}_2$ in the class $\mathcal{I}_\eta$, such that, $k \leq p/4$ and $\gamma \leq \frac{q}{2k}$. For any graph decoder $\psi : \mathcal{X}^{n \times p} \times \mathcal{X}^{n \times p} \to \mathcal{I}_\eta$ that achieves*

$$ \mathsf{P}(\mathcal{I}_\eta^k, d) \leq \delta , \tag{16} $$

*for $d = \theta k$, for some $\theta \in (0, \frac{1}{4})$, the sample size $n$ should satisfy*

$$ n \geq \max\{B_1, B_2\}(1 - \delta - o(1)) , \tag{17} $$

*where we have defined*

$$ B_1 \triangleq \frac{2(1 - \gamma)\log\bar{q} + \gamma\log q - 2\theta\log p}{\lambda\tanh\lambda} , \tag{18} $$

$$ B_2 \triangleq \frac{(1 - \gamma/2)\log 2 - h(\theta)}{3\lambda k\left(\gamma^2\exp(-\lambda(\sqrt{\gamma k}) - 1)/2) + \bar{\gamma}^2\exp(-\lambda(\sqrt{\bar{\gamma}k} - 1)/2)\right)} . \tag{19} $$

Next, we discuss the different scaling behavior of the necessary conditions on sample complexity from Theorem 2. Note that $B_1$ and $B_2$ have different scaling behavior in terms of $\lambda$, $k$ and $p$. In all the following regimes, we assume that $k$ is increasing with $p$.

1. $\lambda = \Theta(1)$: In this regime, $B_1$ scales as $\log p$ and $B_2$ scales as $e^{\sqrt{k}}$. Clearly, $B_2$ dominates the lower bound on the sample complexity if $k = \omega(\log(\log p))$.

2. $\lambda = O(\sqrt{k^{-1}})$: By noting that $\tanh(\lambda) = O(\lambda)$ we find that $B_1$ scales as $\Omega(k \log p)$. $B_2$ scales as $1/k$. Clearly, the sample complexity is dominated by $B_1$.

3. $\lambda = \Theta(\sqrt{k})$ : In this regime, $B_1$ scales as $O(\frac{\log p}{\lambda\tanh\lambda})$ and $B_2$ scales as $e^{k^{1.5}}/k^{1.5}$. If we have $k = \omega(\log(\log p))$, $B_2$ dominates the sample complexity and scales exponentially in $k^{1.5}$.

**Theorem 3** (Class $\mathcal{I}_\eta$ with $k = \Omega(p)$). *Consider a pair of $\eta-$ similar graphs $\mathcal{G}_1$ and $\mathcal{G}_2$ in the class $\mathcal{I}_\eta$, such that, $k = \lfloor cp^{1+\mu} \rfloor$ for given constants $c > 0$ and $\mu \in [0, 1)$, and $\gamma \leq \min\{\eta, \frac{\eta^2 p^2}{4k}\}$. For any graph decoder $\psi : \mathcal{X}^{n \times p} \times \mathcal{X}^{n \times p} \to \mathcal{I}_\eta^k$ that achieves*

$$\mathsf{P}(\mathcal{I}_\eta, d) \leq \delta \ , \tag{20}$$

*for $d = \theta k$ where $\theta \in (0, \frac{1}{4})$, the sample size $n$ should satisfy*

$$n \geq \max\{B_3, B_2\}(1 - \delta - o(1)) \ , \tag{21}$$

*where we have defined $B_2$ in* (19)*, and*

$$B_3 \triangleq [(1 - \gamma/2)\log 2 - h(\theta)] \cdot \lambda^{-1} \frac{\exp(2\lambda)\cosh(4\lambda cp^\mu) + 1}{\exp(2\lambda)\cosh(4\lambda cp^\mu) - 1} \ . \tag{22}$$

To analyze the asymptotic scaling behavior of the necessary condition, we note that $B_2$ depends on $\lambda$ and $k$, and $B_3$ depends on $\lambda$ and $p$. Therefore, depending on the variations of $\lambda$ with respect to $p$ and $k$, we characterize the scaling behavior of the sufficient condition in terms of $k$ and $p$. In the following regimes, we assume that $k$ is increasing with $p$.

1. $\lambda = \Theta(1)$: In this regime, $B_2$ scales as $\exp(\sqrt{k})$. On the other hand, we have $\frac{\exp(2\lambda)\cosh(4\lambda cp^\mu) - 1}{\exp(2\lambda)\cosh(4\lambda cp^\mu) + 1} = \Theta(1)$ and therefore, $B_3 = \Theta(1)$ as $p \to \infty$. Clearly, $B_2$ dominates the bound on sample complexity.

2. $\lambda = O(\sqrt{k^{-1}})$: In this regime, $B_2$ scales as $1/k$. The analysis of $B_3$ shows that $\frac{\exp(2\lambda)\cosh(4\lambda cp^\mu) - 1}{\exp(2\lambda)\cosh(4\lambda cp^\mu) + 1} = O(\max\{1/\sqrt{k}, k/p^2\})$. Therefore, $B_3$ scales according to $\Omega(\min\{k, p^2/\sqrt{k}\})$. Note that when we have $k = \Omega(p)$ and $k = O(p^{4/3})$, we have $\min\{k, p^2/\sqrt{k}\} = k$ and therefore, the bound on sample complexity scales as $\Omega(k)$. When $k = \Omega(p^{4/3})$, the bound on the sample complexity scales as $\Omega(p^2/\sqrt{k})$ asymptotically.

3. $\lambda = \Theta(\sqrt{k})$: In this regime, $B_2$ scales as $e^{k^{1.5}}$ and $B_3 \to 0$ as $k \to \infty$. Therefore, the lower bound on sample complexity scales exponentially in $k^{1.5}$.

The analysis of the results in Theorem 1 and Theorem 2 reveals that the sufficient and the necessary bounds on the sample complexity scale at the same rate (non-exponential) for the class $\mathcal{I}_\eta$ under a particular regime, as described in Corollary 1.

**Corollary 1** (Optimal Sample Complexity). *When the maximum number of edges is fixed and satisfies $k \leq p/4$, and we have*

$$\gamma \leq \min\left\{\frac{q}{2k}, \frac{\eta^2 p^2}{4k}\right\} \qquad and \qquad \lambda = O(1/p) \ , \tag{23}$$

*Theorem 1 indicates that when $n > c_2 p^2 \log p$, for a constant $c_2$, there exists a graph decoder that recovers both graphs with $\mathsf{P}(\mathcal{I}_\eta, d) \leq \delta$. On the other hand, in this regime, Theorem 2 indicates that for any graph decoder to achieve $\mathsf{P}(\mathcal{I}_\eta, d) \leq \delta$ we should have $n > c_3 p^2 \log p$, for some constant $c_3 > 0$. Therefore, in this regime, the graph decoder that satisfies Theorem 1 achieves the optimal sample complexity up to constant factors.*

Furthermore, we comment that the extreme case of $\eta = 0$ corresponds to recovering two independent graphs the other extreme case of $\eta = 1$ corresponds to recovering two identical graphs. In both these extreme cases, the problem analyzed in this paper simplifies to the problem of structure learning of one graph studied in Scarlett and Cevher [2016] (for approximate recovery) and in Santhanam and Wainwright [2012] (for exact recovery, i.e., $d = 0$). In general, however, when we depart from these special cases, the analysis techniques in the context of single graphs in existing literature do not extend directly to the context of recovering a pair of graphs with structural similarity. Specifically, we use novel ensemble constructions for a pair of graphs that accommodate structural similarity in different regimes of $k$ and analyze the pairwise KL divergences for the graph pairs to recover the necessary conditions in Theorems 2 and 3. Moreover, our analysis of an ML decoder also recovers the hitherto uninvestigated sufficient conditions for the sample complexity under the approximate recovery criterion. Hence, the results provided in this paper are completely different. This observation is formalized in the following corollary.

**Corollary 2** (Special Cases). *The necessary and sufficient conditions on sample complexity in the extreme cases of $\eta = 0$ and $\eta = 1$ subsume the existing results for structure learning in single graphs.*

In the context of the asymptotic scaling behaviors summarized in Table 1, we comment that the necessary condition bounds for approximate recovery are not any looser with respect to that for exact recovery than those in the context of single-graph recovery. Also, in some regimes the gap between the necessary conditions and the sufficient conditions on the sample complexity is tighter than others. For instance, when $k = O(p)$ the mismatch is only a factor $k$. The mismatch between the necessary conditions and sufficient conditions is more profound in denser graphs, i.e., when $k = \Omega(p)$.

Furthermore, the analysis of the necessary conditions in Theorems 2 an 3 and sufficient conditions in Theorem 1 reveals that $d$ does not affect the asymptotic scaling rate of their respective bounds even when $d$ scales as fast as linearly with $k$. For instance, in Theorem 1, $d$ appears only in a logarithmic factor scaling at most at the rate of $\log k$ which is dominated by $k \log p$ in $A_1$ and $A_2$. Therefore, the results in Table 1 do not depend on $d$.

## 5   Numerical Evaluations

In this section, we evaluate the tradeoffs between decision reliability captured by $\mathsf{P}(\mathcal{I}_\eta, d)$ defined in (7) and the necessary and sufficient conditions on the sample complexities established. We evaluated these tradeoffs for different approximate recovery levels controlled by $d$ as well as similarity levels of the two graphs specified by $\eta$. In general, the implementation of an ML decoder may become infeasible as the size of the graphs grow. Therefore, to gain meaningful insights in the sample complexity with increasing size of graphs, the evaluations were performed on an ensemble of graphs that contains graphs with many isolated edges. In this ensemble, we set the size of the graphs to $p$, with $q = \lfloor \eta p \rfloor$ nodes in the shared subgraph. We assumed that each graph contains $\alpha$ isolated edges, with $\lfloor \eta\alpha \rfloor$ edges lying in the shared subgraph. Furthermore, the graphs in this ensemble were constructed in the following manner. We grouped the non-shared cluster with size $(p - q)$ vertices in $(p - q)/2$ fixed pairs and randomly connected the vertices in $(\alpha - \lfloor \eta\alpha \rfloor)$. Similarly, the $q$ vertices of the shared subgraph were grouped into $q/2$ fixed pairs and $\lfloor \eta\alpha \rfloor$ pairs were selected randomly to be connected. For this ensemble, the implementation of ML decoder can be readily shown to be equivalent to a counting scheme that counts the number of agreements in the states of different nodes in the data. This allowed us to visualize the behavior of the sample complexity for ML decoder as the size of the graphs was increased.

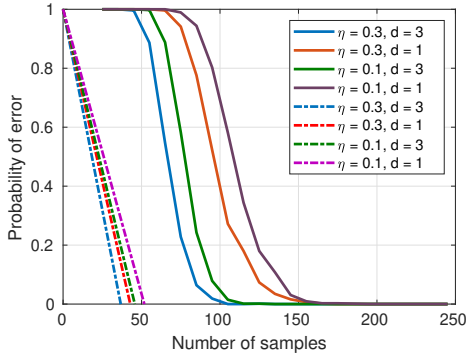

Figure 2: Reliability ($\mathsf{P}(\mathcal{I}_\eta, d)$) versus sample complexity ($n$) for different values of $\eta$ and $d$. Solid and dashed curves represent the sufficient number of samples (based on ML decoder) and necessary number of samples, respectively.

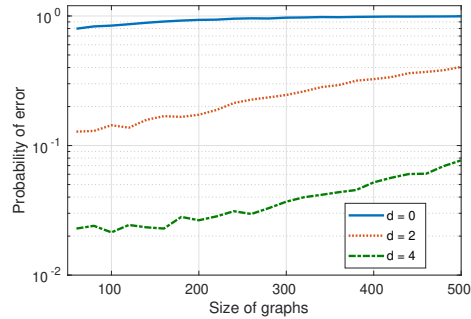

Figure 3: Reliability ($\mathsf{P}(\mathcal{I}_\eta, d)$) versus graph size ($p$) for different values of $d$, where $d$ represents the tolerance to distortion in the recovered graphs with respect to the true graphs.

We first considered a graph with $p = 100$ vertices and $\alpha = 20$. Figure 2 depicts the variations of the error probability $\mathsf{P}(\mathcal{I}_\eta, d)$ versus $n$. For each value of $\mathsf{P}(\mathcal{I}_\eta, d)$, the figure specifies the necessary (shown by dashed curves) and sufficient conditions (shown by solid curves) on the number of samples $n$. The sufficient conditions are obtained by simulations of the ML decoder. The figure shows

these variations for different levels of graph similarity $\eta = 0.1, 0.3$ and different values of recovery approximation $d = 1, 3$. The probability of error was evaluated empirically over 6000 trials.

In Corollary 1, we have provided a regime in which the scaling behaviors of the necessary and sufficient conditions on the sample complexity coincide, establishing the exact sample complexity. In this regime, as a result, the ML decoder achieves an optimal structure learning rule. We used the ML rule to characterize the variations of decision reliability $\mathsf{P}(\mathcal{I}_\eta, d)$ as the size of the graph varied in the range $p \in [50, 500]$ for fixed number of edges. Figure 3 depicts these variations. For the results in this figure we have fixed $\alpha = 20$ and $\eta = 0.5$, and have evaluated the performance based on $n = 40$ samples from each graph.

In Fig. 3, we observe that the decision reliability measure $\mathsf{P}(\mathcal{I}_\eta, d)$ achieves lower error rate with increase in $d$ for the same number of samples. It is important to note that increase in $d$ signifies a rise in tolerance to errors in the structure recovery by the graph decoder, and therefore, the decline in quality of structure recovery decisions with respect to the ground truth. Also, as stated in Corollary 1, we observe that graph recovery becomes more difficult as the graph size increases while $k$ remains fixed.

## 6   Conclusion

In this paper, we have considered the problem of structure learning in the presence of side information about the structure. This is posed, naturally, as jointly recovering the structures of two graphs with partial internal structural similarities. Specifically, it is assumed that both graphs share an identical subgraph. Any inference about the structure of this subgraph from either of the graphs serves as the side information for recovering the structure of the other graph. A general recovery criterion that encompasses both exact and partial recovery of the graphs is considered. We have established necessary (information-theoretic) and sufficient (algorithmic) bounds on the sample complexity for achieving a bounded probability of error in structure recovery. The scaling behaviors of these conditions are analyzed in different regimes. We have also identified a regime in which the necessary and sufficient conditions coincide, establishing the optimal sample complexity. We have also provided numerical evaluations to illustrate the interplay among the various parameters involved.

The setting studied in this paper has been motivated from applications in a broad range of domains like social networks, genetics, and behavioral analysis. While the existing works have primarily focused on context specific algorithmic frameworks for joint inference, our results have established the information-theoretic benchmarks on the sample complexity in different regimes characterized by the properties of the graph structures.

## Footnotes

[1]The results in this paper can be generalized to settings with more than two graphs. For clarity, we analyze the setting with two graphs.

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
