[Supplementary Material]

# A    Proofs of Sufficient Conditions

We analyze the sample complexity of an ML based decoder using the large deviations bound. Consider a class $\mathcal{I}_\eta$ consisting of $M = |\mathcal{I}_\eta|$ pair of graphs and let the index $i \in \{1, \ldots, M\}$ be associated with a graphical model consisting of a pair of graphs $(\mathcal{G}_1^i, \mathcal{G}_2^i) \in \mathcal{I}_\eta$, where the model $i$ is characterized by the parameters $\theta^i = \{\theta_1^i, \theta_2^i\}$ where $\theta_m^i \in \mathbb{R}^{\binom{p}{2}}$ is the set of edges for $\mathcal{G}_m$ under model $i \in \{1, \ldots, M\}$. We denote the set of all possible edges in the pair of graphs by $\Theta$.

Given the two collections of graph samples $(\mathbf{X}_1^n, \mathbf{X}_2^n)$, the ML decoder decides on the true edge structures according to the rule given by

$$(\hat{\mathcal{G}}_1, \hat{\mathcal{G}}_2) = \arg \max_{\theta_i \in \Theta} \ell_{\theta^i}(\mathbf{X}_1^n, \mathbf{X}_2^n) , \tag{24}$$

where $\ell_{\theta^i}(\mathbf{X}_1^n, \mathbf{X}_2^n) \triangleq f_{12}^i(\mathbf{X}_1^n, \mathbf{X}_2^n)$ is the log likelihood with respect to the model $i \in \{1, \ldots, M\}$ and $f_{12}^i(\cdot)$ is the joint pdf of the samples collected from the two graphs under the model $i \in \{1, \ldots, M\}$.

Since given the true model pair, the samples $X_1$ and $X_2$ are generated independently for both graphs, we have

$$f_{12}^i(X_1, X_2) = f_1^i(X_1)f_2^i(X_2) , \tag{25}$$

If the maximum in (24) is not unique, we randomly select the model from the set of models corresponding to the maximum likelihood. If the data $(\mathbf{X}_1^n, \mathbf{X}_2^n)$ is collected from a pair of graphs with true model $\theta^i$, the ML decoder fails when there exists some other model $\theta^j$, $j \neq i$, s.t., $\ell_{\theta^j}(\mathbf{X}_1^n, \mathbf{X}_2^n) \geq \ell_{\theta^i}(\mathbf{X}_1^n, \mathbf{X}_2^n)$ and at least one of the estimated edge structures has an edit distance greater than $d$ with respect to its respective true graph. This event is represented by $\theta^j \Delta \theta^i > d$. Therefore, we have

$$\mathbb{P}[\theta^j \Delta \theta^i > d]$$

$$= \mathbb{P}\left( \bigcup_{\substack{\theta^j \in \Theta \backslash \theta^i \\ \theta^j \Delta \theta^i > d}} \ell_{\theta^j}(\mathbf{X}_1^n, \mathbf{X}_2^n) \geq \ell_{\theta^i}(\mathbf{X}_1^n, \mathbf{X}_2^n) \right) \tag{26}$$

$$\leq \sum_{\substack{\theta^j \in \Theta \backslash \theta^i \\ \theta^j \Delta \theta^i > d}} \mathbb{P}[\ell_{\theta^j}(\mathbf{X}_1^n, \mathbf{X}_2^n) \geq \ell_{\theta^i}(\mathbf{X}_1^n, \mathbf{X}_2^n)] , \tag{27}$$

where (27) follows from the union bound. Next, we provide the large deviations bound in Lemma 1 that provides the sufficient conditions for the probability of error of the ML decoder to vanish.

**Lemma 1.** *Given the i.i.d graph samples $(\mathbf{X}_1^n, \mathbf{X}_2^n)$ from the model $\theta^i \in \Theta$, for any model $j \neq i$, we have*

$$\mathbb{P}[\ell_{\theta^j}(\mathbf{X}_1^n, \mathbf{X}_2^n) \geq \ell_{\theta^i}(\mathbf{X}_1^n, \mathbf{X}_2^n)]$$

$$\leq \exp(-\frac{n}{2}(J(\theta_1^i \| \theta_1^j) + J(\theta_2^i \| \theta_2^j))) , \tag{28}$$

*where*

$$J(\theta_m^i \| \theta_m^j) \triangleq D_{\mathsf{KL}}\left( \frac{\theta_m^i + \theta_m^j}{2} \| \theta_m^i \right)$$

$$+ D_{\mathsf{KL}}\left( \frac{\theta_m^i + \theta_m^j}{2} \| \theta_m^j \right) , \tag{29}$$

*for $m \in \{1, 2\}$.*

*Proof.* Let $R \triangleq \ell_{\theta^j}(\mathbf{X}_1^n, \mathbf{X}_2^n) - \ell_{\theta^i}(\mathbf{X}_1^n, \mathbf{X}_2^n)$. Then, using Chernoff's bound, we have

$$\mathbb{P}(R \geq 0) \leq \min_{s > 0} \mathbb{E}_i[\exp(sR)] . \tag{30}$$

Note that

$$\mathbb{E}_i[\exp(sR)]$$

$$= \sum_{\mathbf{X}_1^n, \mathbf{X}_2^n} \left( \exp\left( \sum_{w=1}^n s\ell_{\theta^j}(X_1(w), X_2(w)) \right. \right.$$

$$\left. \left. - s\ell_{\theta^i}(X_1(w), X_2(w))) \right) \right)$$

$$\times \prod_{m=1}^n f_{\theta^i}(X_1(m), X_2(m)) , \tag{31}$$

where $X_u(w)$ is the $w-$th sample of $\mathbf{X}_u^n$ and $f_{\theta^i}$ is the joint distribution of one pair of graph samples under the model $\theta^i$. Then, using

$$\ell_{\theta^i}(X_1(w), X_2(w)) = \log f_{\theta^i}(X_1(w), X_2(w)) , \tag{32}$$

and from (31), we have

$$\mathbb{E}_i[\exp(sR)]$$

$$= \sum_{\mathbf{X}_1^n, \mathbf{X}_2^n} \prod_{w=1}^n [f_{\theta^j}(X_1(w), X_2(w))]^s$$

$$\times [f_{\theta^i}(X_1(w), X_2(w))]^{1-s} , \tag{33}$$

$$= \left( \sum_{X_1, X_2} [f_{\theta^j}(X_1, X_2)]^s [f_{\theta^i}(X_1, X_2)]^{1-s} \right)^n . \tag{34}$$

Using (34) and (25), we have

$$\mathbb{E}_i[\exp(sR)]$$

$$= \left( \sum_{X_1} \left[ [f_1^j(X_1)]^s [f_1^i(X_1)]^{1-s} \right] \right.$$

$$\left. \times \sum_{X_2} \left[ [f_1^j(X_2)]^s [f_1^i(X_2)]^{1-s} \right] \right)^n . \tag{35}$$

When $s = 1/2$, and using the expansions of $f_1^i$ and $f_1^j$, it can be readily verified that

$$\sum_{X_1 \in \{-1,1\}^p} \left[ [f_1^j(X_1)]^{1/2} [f_1^i(X_1)]^{1/2} \right] = \frac{Z_1(\frac{\theta^i + \theta^j}{2})}{(Z_1(\theta^i) Z_1(\theta^j))^{1/2}} , \tag{36}$$

$$= \exp\left(-\frac{J(\theta_1^i \| \theta_1^j)}{2}\right) , \tag{37}$$

where $J(\theta_1^i \| \theta_1^j)$ is defined in (29). Following the similar analysis as in (36) and (37) for $\mathcal{G}_2$, and by setting $s = 1/2$ in (34), we have

$$\mathbb{E}_i[\exp(R/2)] = \exp\left(-\frac{n}{2}(J(\theta_1^i \| \theta_1^j) + J(\theta_2^i \| \theta_2^j))\right) \tag{38}$$

From (30) and (38), the proof of Lemma 1 is completed. $\qquad \square$

Next, we use [Santhanam and Wainwright, 2012, Lemma 4] to find a lower bound on the divergence $J(\theta_1^i \| \theta_1^j) + J(\theta_2^i \| \theta_2^j))$ in terms of the edge mismatch between the models $i$ and $j$. Define $T(\theta_m^i, \theta_m^j)$ as the matching number of the graph whose edges are given by the set

$$E_m^i \triangle E_m^j \triangleq (E_m^i \setminus E_m^j) \cup (E_m^i \setminus E_m^j) , \tag{39}$$

where $E_m^i$ is the set of edges for graph $\mathcal{G}_m$ under model $i \in \{1, \ldots, M\}$. $E_m^i \triangle E_m^j$ is referred to as *edit distance* in existing literature. Then, using [Santhanam and Wainwright, 2012, Lemma 4], we have

$$
\begin{aligned}
J(\theta_1^i \| \theta_1^j) + J(\theta_2^i \| \theta_2^j) & \\
&\geq \frac{T(\theta_1^i, \theta_1^j) + T(\theta_2^i, \theta_2^j)}{3\zeta^2 + 1} \sinh^2\left(\frac{\lambda}{4}\right) ,
\end{aligned}
\tag{40}
$$

where $\log \zeta$ is the maximum neighborhood weight.

## A.1 Proof of Theorem 1

Consider the models $i$ and $j$ in the class $\mathcal{I}_\eta$, s.t., the non-shared parts of the graphs $\mathcal{G}_1^i$ and $\mathcal{G}_1^j$ differ in $\ell_1$ edges, that of $\mathcal{G}_2^i$ and $\mathcal{G}_2^j$ differ in $\ell_2$ edges, and the shared part of the two models differ in $\ell_{\sf s}$ number of edges. Therefore, $\ell_u \in \{0, \ldots, \min\{2k, 2\binom{p-\lfloor \eta p \rfloor}{2} + 2p\lfloor \eta p \rfloor\}$, for $u \in \{1, 2\}$, and $\ell_{\sf s} \in \{0, \ldots, 2\lfloor \gamma k \rfloor\}$. For the sake of clarity, we define

$$
k' \triangleq \min\left\{ k, \binom{p - \lfloor \eta p \rfloor}{2} + (p - \lfloor \eta p \rfloor)\lfloor \eta p \rfloor \right\} .
\tag{41}
$$

By using notion of vertex cover, we conclude that there are at most $2^{2k'} p^{2(\ell_1 + \ell_2)(k'+1)} \times 2^{\lfloor \gamma k \rfloor} \lfloor \eta p \rfloor^{2\ell_{\sf s}(\lfloor \gamma k \rfloor + 1)}$ number of models that differ in $\ell_1$ edges in the non shared part of $\mathcal{G}_1^i$, $\ell_2$ edges in the non shared part of $\mathcal{G}_2^{(j)}$, and $\ell_{\sf s}$ edges in the shared part of model $i$. Using (27), the large deviations bound in Lemma 1, and (40), we obtain

$$
\begin{aligned}
\mathbb{P}&[\theta^j \Delta \theta^i > d] \\
&\leq \sum_{\substack{\ell_1 + \ell_s > d \\ \text{or} \\ \ell_2 + \ell_s > d}} \sum_{\ell_1 = 0}^{2k'} \sum_{\ell_2 = 0}^{2k'} \sum_{\ell_{\sf s} = 0}^{2\lfloor \gamma k \rfloor} 2^{2k' + \lfloor \gamma k \rfloor} \Big( p^{2(\ell_1 + \ell_2)(k'+1)} \\
&\quad \times \lfloor \eta p \rfloor^{2\ell_{\sf s}(\lfloor \gamma k \rfloor + 1)} \\
&\quad \times \exp\big(-n \frac{\ell_1 + \ell_2 + 2\ell_{\sf s}}{3\zeta^2 + 1} \sinh^2\left(\frac{\lambda}{4}\right)\big)\Big) \\
&\leq \sum_{\ell_s = d+1}^{2\lfloor \gamma k \rfloor} 2^{2k' + \lfloor \gamma k \rfloor} \lfloor \eta p \rfloor^{2\ell_s(\lfloor \gamma k \rfloor + 1)} \exp\big(-n \frac{2\ell_s}{3\zeta^2 + 1} \sinh^2\left(\frac{\lambda}{4}\right)\big) + \\
&\quad 2 \max_{\ell_s \in \{0, \ldots, d\}} \sum_{\ell_1 = d+1-\ell_s} 2^{2k' + \lfloor \gamma k \rfloor} p^{2\ell_1(k'+1)} \exp\big(-n \frac{\ell_1}{3\zeta^2 + 1} \sinh^2\left(\frac{\lambda}{4}\right)\big) .
\end{aligned}
$$

(42)

(43)

We define

$$
C(\ell_1) \triangleq p^{2\ell_1(k'+1)} \exp\big(-n \frac{\ell_1}{3\zeta^2 + 1} \sinh^2\left(\frac{\lambda}{4}\right)\big)
\tag{44}
$$

and

$$
D(\ell_{\sf s}) \triangleq \lfloor \eta p \rfloor^{2\ell_{\sf s}(\lfloor \gamma k \rfloor + 1)} \exp\big(-n \frac{2\ell_{\sf s}}{3\zeta^2 + 1} \sinh^2\left(\frac{\lambda}{4}\right)\big) .
\tag{45}
$$

We simplify (43) to

$$
\begin{aligned}
\mathbb{P}&[\theta^j \Delta \theta^i > d] \\
&\leq 2^{2k' + \lfloor \gamma k \rfloor} \left( \sum_{\ell_s = d+1}^{2\lfloor \gamma k \rfloor} D(\ell_{\sf s}) + 2 \max_{\ell_s \in \{0, \ldots, d\}} \sum_{\ell_1 = d+1-\ell_s}^{2k'} C(\ell_1) \right) .
\end{aligned}
\tag{46}
$$

If we have

$$
2^{2k' + \lfloor \gamma k \rfloor + 1} \max_{\ell_s \in \{0, \ldots, d\}} \sum_{\ell_1 = d+1-\ell_s}^{2k'} C(\ell_1) \leq \frac{\delta}{2} ,
\tag{47}
$$

and

$$2^{2k'+\lfloor \gamma k \rfloor} \sum_{\ell_s=d+1}^{2\lfloor \gamma k \rfloor} D(\ell_s) \leq \frac{\delta}{2} \ , \tag{48}$$

then the probability $\mathbb{P}[\theta^j \Delta \theta^i > d] \leq \delta \in (0,1)$. To ensure that (47) is satisfied, we obtain

$$2^{2k'+\lfloor \gamma k \rfloor+1} \max_{\ell_s \in \{0,\ldots,d\}} \sum_{\ell_1=d+1-\ell_s}^{2k'} C(\ell_1)$$

$$\leq \max_{\ell_s \in \{0,\ldots,d\}} \max_{\ell_1 \in \{d+1-\ell_s,\ldots,2k'\}} \exp \Big( (2k' + \lfloor \gamma k \rfloor) + \log(2k' - d - 1 - \ell_s)$$

$$+ 2\ell_1(k'+1)\log p - n\frac{\ell_1}{3\zeta^2+1}\sinh^2\left(\frac{\lambda}{4}\right) \Big) \ , \tag{49}$$

which is less than $\delta/2$ if

$$n \geq \frac{3\zeta^2+1}{\sinh^2(\lambda/4)} \Big( (2k' + \lfloor \gamma k \rfloor) + \log(2k' - d)$$

$$+2(k'+1)\log p + \log \frac{4}{\delta} \Big) \ . \tag{50}$$

To ensure that (48) is satisfied, we obtain

$$2^{2k'+\lfloor \gamma k \rfloor} \sum_{\ell_s=d+1}^{2\lfloor \gamma k \rfloor} D(\ell_s)$$

$$\leq \max_{\ell_s \in \{d+1,\ldots,\lfloor \gamma k \rfloor\}} \exp \Big( (2k' + \lfloor \gamma k \rfloor) + \log(2\lfloor \gamma k \rfloor - d)$$

$$+ 2\ell_s(\lfloor \gamma k \rfloor + 1)\log\lfloor \eta p \rfloor - n\frac{2\ell_s}{3\zeta^2+1}\sinh^2\left(\frac{\lambda}{4}\right) \Big) \ , \tag{51}$$

which is less than $\delta/2$ if

$$n \geq \frac{3\zeta^2+1}{2\sinh^2(\lambda/4)} \Big( (2k' + \lfloor \gamma k \rfloor) + \log(2\lfloor \gamma k \rfloor - d)$$

$$+2(\lfloor \gamma k \rfloor + 1)\log\lfloor \eta p \rfloor + \log \frac{2}{\delta} \Big) \ . \tag{52}$$

For sufficiently large $p$, the conditions on $n$ in Theorem 1 satisfy (47) and (48).

## B   Proofs of Necessary Conditions

### B.1   Fano's Lemma

We state the Fano's Lemma for approximate recovery in the context of joint selection of graphical models.

**Lemma 2.** *Consider a restricted ensemble of graph pairs $\mathcal{T} \subseteq \mathcal{I}_\eta$ with $M = |\mathcal{T}|$, for which the graph decoder's outputs lie in some class $\mathcal{T}'$, without loss of optimality. If there exists a pair of graphs $(\mathcal{G}_1', \mathcal{G}_2') \in \mathcal{T}'$ for each graph pair $(\mathcal{G}_1, \mathcal{G}_2) \in \mathcal{T}$ such that $D_{\mathrm{KL}}(f_{\mathcal{G}_1,\mathcal{G}_2} || f_{\mathcal{G}_1',\mathcal{G}_2'}) \leq \epsilon$, and there are at most $A(d)$ pairs of graphs in $\mathcal{T}'$ within $d$ level of distortion with respect to any pair of graphs $(\mathcal{G}_1, \mathcal{G}_2) \in \mathcal{T}$, then $\mathsf{P}(\mathcal{I}_\eta, q_{\mathsf{max}}) \leq \delta$ if*

$$n \geq \frac{\log M - \log A(d)}{\epsilon} \left( 1 - \delta - \frac{\log 2}{\log M} \right) \tag{53}$$

## B.2 Ensemble Constructions and Proofs for Approximate Recovery

- *Ensemble* 1: We use this ensemble to derive the bound $B_1$. In this ensemble, we consider a set of graph pairs such that each graph consists of exactly $\alpha \leq \min\{\eta p, (1-\eta)p\}/4$ number of node-disjoint edges, with $\lfloor \gamma k \rfloor$ edges in the shared cluster of the graph. Therefore, the number of such graph pairs are given by

$$|\mathcal{T}| = \prod_{i=0}^{\lfloor \gamma \alpha \rfloor} \binom{\eta p - 2i}{2} \left( \prod_{j=0}^{\alpha - \lfloor \gamma \alpha \rfloor} \binom{p - \eta p - 2j}{2} \right)^2 \tag{54}$$

$$\geq \binom{\lfloor \eta p/2 \rfloor}{2}^{\lfloor \gamma \alpha \rfloor} \binom{\lfloor (p - \eta p)/2 \rfloor}{2}^{2(\alpha - \lfloor \gamma \alpha \rfloor)} . \tag{55}$$

Since all the edges are node-disjoint, the total number of nodes used in the shared cluster is given by $2\lfloor \gamma k \rfloor \leq \lfloor \eta p \rfloor$ if

$$\gamma \leq \frac{\lfloor \eta p \rfloor}{2k} . \tag{56}$$

Let the output of the graph decoder, represented by the set $\mathcal{T}'$, be the set of all possible graph pairs. Then, the number of graph pairs that lie within the distortion level $d$ of any given graph pair in this ensemble is bounded as

$$A(d) \leq \left( \sum_{q_1=0}^{d} \sum_{q_1'=0}^{d-q_1} \binom{\alpha}{q} \binom{p}{2}^{q_1'} \right)^2$$

$$\leq (1+d)^2 \binom{\alpha}{\lfloor \alpha/2 \rfloor}^2 \binom{p}{2}^{2d} , \tag{57}$$

where $\binom{\alpha}{q_1}$ represents the number of ways to remove $q_1$ edges from the base graph and $\binom{p}{2}^{q_1'}$ is the upper bound on the number of ways to add $q_1'$ number of edges to the base graph. It has been shown in multiple existing studies in the context of structure learning of Ising models that the KL divergence between any two graphs with isolated edges is upper bounded by $\lambda \tanh \lambda$. We can readily verify that the KL divergence between any two pair of graphs in this ensemble is upper bounded by $2\alpha\lambda \tanh \lambda$. Using Lemma 2, we get the condition that the error probability in approximately recovering any graph pair in the class $\mathcal{I}_\eta$, $\mathsf{P}_{\mathcal{I}_\eta} \leq \delta$ if

$$n \geq \left( \frac{\lfloor \gamma \alpha \rfloor \log \binom{\lfloor \eta p/2 \rfloor}{2} + 2(\alpha - \lfloor \gamma \alpha \rfloor) \log(\binom{\lfloor (p - \eta p)/2 \rfloor}{2})}{2\alpha\lambda \tanh \lambda} \right.$$

$$\left. - \frac{2\log \left( (1+d)^2 \binom{\alpha}{\lfloor \alpha/2 \rfloor} \binom{p}{2}^d \right)}{2\alpha\lambda \tanh \lambda} \right) \left( 1 - \delta - \frac{\log 2}{|\mathcal{T}|} \right) \tag{58}$$

We use the simplification that $\log \binom{p}{2} \approx (2\log p)(1 + o(1))$, $\binom{\lfloor (p - \eta p)/2 \rfloor}{2} \approx 2\log(p - \eta p)(1 + o(1))$, and $\binom{\lfloor \eta p/2 \rfloor}{2} \approx 2\log(\eta p)(1 + o(1))$. Also, note that $\lfloor \gamma \alpha \rfloor \in [\gamma\alpha - 1, \gamma\alpha + 1/2]$. Also, $\log \binom{\alpha}{\lfloor \alpha/2 \rfloor} \leq \alpha \log 2 = o(\alpha \log p)$ and if $d \leq (1 - \Omega(1))\alpha$, $\log(1 + d)^2 \leq 2\log(1 + \alpha) = o(\alpha \log p)$. Using these facts, we can simplify the bound in (58) to

$$n \geq \frac{2\gamma\alpha \log \eta p + 4(\alpha - \gamma\alpha) \log(p - \eta p) - 4d \log p}{2\alpha\lambda \tanh \lambda}$$

$$\times (1 - \delta - o(1)) , \tag{59}$$

when $\alpha \to \infty$. When $d = \lfloor \theta \alpha \rfloor$ for some $\theta \in (0, 1)$, (59) is simplified to

$$n \geq \frac{\gamma \log \eta p + 2(1 - \gamma) \log(p - \eta p) - 2\theta \log p}{\lambda \tanh \lambda}$$

$$\times (1 - \delta - o(1)) . \tag{60}$$

- *Ensemble* 2: We use this construction to derive the bounds $B_3$ and $B_5$. In this ensemble, we divide the nodes in the shared part of the graph pair into $\alpha_1$ fixed groups and the non-shared part of each graph is divided into $\alpha_2$ groups, with each group containing $m$ vertices. Therefore, the total number of vertices used in each graph is $m(\alpha_1 + \alpha_2)$ and under the assumption that there are no inter-group edges, the total number of possible edges in a graph is $2^{(\alpha_1+\alpha_2)\binom{m}{2}}$. Considering the $\eta-$similarity between the graphs, the total number of graph pairs is $|\mathcal{T}| = 2^{(\alpha_1+2\alpha_2)\binom{m}{2}}$. Note that the maximal degree of any graph in this ensemble is $m-1$. The number of graph pairs that lie within $d$ level of distortion of any true graph pair is bounded as

$$
\begin{aligned}
A(d) &\leq \left( \sum_{q=0}^{d} \binom{(\alpha_1+\alpha_2)\binom{m}{2}}{q} \right)^2 \\
&\leq (1 + d\binom{(\alpha_1+\alpha_2)\binom{m}{2}}{d})^2 ,
\end{aligned}
\tag{61}
$$

if $d \leq \frac{1}{2}(\alpha_1+\alpha_2)\binom{m}{2}$. The KL divergence between any two sets of graph-pairs is upper bounded by $\epsilon = 2(\alpha_1+\alpha_2)\binom{m}{2}\lambda\frac{e^{2\lambda}\cosh(2\lambda m)-1}{e^{2\lambda}\cosh(2\lambda m)+1}$, where the proof follows directly from the application of [Scarlett and Cevher, 2016, Lemma 4]. Setting $d = \lfloor \theta(\alpha_1+\alpha_2)\binom{m}{2}\rfloor$ and using Lemma 2 and the identity

$$
\binom{(\alpha_1+\alpha_2)\binom{m}{2}}{\lfloor \theta(\alpha_1+\alpha_2)\binom{m}{2}\rfloor} = e^{((\alpha_1+\alpha_2)\binom{m}{2}h(\theta)(1+o(1))},
\tag{62}
$$

when $(\alpha_1+\alpha_2)\binom{m}{2} \to \infty$ , we get

$$
\begin{aligned}
n &\geq \frac{(\alpha_1+2\alpha_2)\log 2 - 2(\alpha_1+\alpha_2)h(\theta)}{2(\alpha_1+\alpha_2)\lambda\frac{e^{2\lambda}\cosh(2\lambda m)-1}{e^{2\lambda}\cosh(2\lambda m)+1}} \\
&\times (1 - \delta - o(1)) .
\end{aligned}
\tag{63}
$$

Next, we provide appropriate choices for $m$, $\alpha_1$ and $\alpha_2$ for application to the classes $\mathcal{I}_k^{\eta,\gamma}$ and $\mathcal{I}_{d,k}^{\eta,\gamma}$. For $\mathcal{I}_\eta$, we set the maximum number of edges in a graph to $k = \lfloor cp^{1+\mu}\rfloor$ for some $c > 0$ and $\mu \in [0,1]$. We choose $m = \lfloor 2cp^\mu\rfloor$, $\alpha_1 = \lfloor \gamma p/m\rfloor$ and $\alpha_2 = \lfloor (1-\gamma)p/m\rfloor$. Note that for these choices, $\alpha_1 = \frac{p^{1-\mu}}{2c}(1+o(1))$ and $\alpha_2 = \frac{(1-\gamma)p^{1-\mu}}{2c}(1+o(1))$. Also, the number of nodes used in the shared cluster for this construction is $\alpha_1 m \leq \gamma p \leq \eta p$ if $\gamma \leq \eta$. Also, the number of possible edges in the shared cluster of the graph-pair is

$$
\alpha_1\binom{m}{2} \leq \frac{\alpha_1 m^2}{2} \leq \frac{\gamma pm}{2} \leq c\gamma p^{1+\mu}
\tag{64}
$$

and that in the non-shared component of a graph is

$$
\alpha_2\binom{m}{2} \leq \frac{\alpha_2 m^2}{2} \leq \frac{(1-\gamma)pm}{2} \leq c(1-\gamma)p^{1+\mu} .
$$

Therefore, the total number of possible edges in this construction is $(\alpha_1+\alpha_2)\binom{m}{2} \leq cp^{1+\mu}$. The use of these specific values of $m$, $\alpha_1$ and $\alpha_2$ in (63) leads to

$$
n \geq \frac{(1-\gamma/2)\log 2 - h(\theta)}{\lambda\frac{e^{2\lambda}\cosh(4\lambda cp^\mu)-1}{e^{2\lambda}\cosh(4\lambda cp^\mu)+1}}(1 - \delta - o(1)) .
\tag{65}
$$

- *Ensemble* 3: We use this ensemble to derive the bounds $B_2$ and $B_4$. Under this ensemble, we first discuss the construction of the base graph, modifications to which leads to other graph-pairs in this ensemble. Consider a group of $2m_1$ vertices that are divided into fully connected sub-groups of $m_1$ vertices. Then, we put $m_1$ edges between the two sets of vertices in an arbitrary fashion. We assume that there are $\alpha_1$ disjoint copies of this group of $2m_1$ vertices in the shared cluster and $\alpha_2$ disjoint copies of a similarly constructed group of $2m_2$ vertices in the non-shared clusters of a graph pair in this ensemble, and the resulting

graph pair is assumed to be the base graph pair. Next, we form each graph pair in this ensemble by adding additional edges arbitrarily in each of the previously constructed groups of $2m_1$ and $2m_2$ sized cliques. Note that these additional edges are identical in the shared cluster of the graph-pair and may be distinct in the non shared clusters of the $\eta-$similar graphs.

Therefore, the number of nodes used in a graph is $2(\alpha_1 m_1 + \alpha_2 m_2)$. The total number of edges in the shared cluster of the graph is upper bounded by $\alpha_1 \binom{2m_1}{2} \leq 2\alpha_1 m_1^2$. Similarly, the total number of edges in the non-shared cluster of the graph is upper bounded by $\alpha_2 \binom{2m_2}{2} \leq 2\alpha_2 m_2^2$. Note that the construction of the base graph consists of $m_1$ edges between the two subgroups in a group of $2m_1$ vertices and $m_2$ edges between the two subgroups in a group of $2m_2$ vertices. Therefore, there can be at most $m_1^2 - m_1$ additional edges in a group of $2m_1$ vertices and $m_2^2 - m_2$ additional edges in a group of $2m_2$ vertices This implies that the total number of graph pairs in this ensemble is given by $2^{(\alpha_1 m_1(m_1-1) + 2\alpha_2 m_2(m_2-1))}$. Also, the maximum degree of any vertex in the shared cluster is $2m_1 - 1$ and that in the non-shared cluster is $2m_2 - 1$. The total number of graph pairs within $d$-distortion of any graph pair is upper bounded by

$$
\begin{aligned}
A(d) &\leq \left( \sum_{q=0}^{d} \binom{\alpha_1 m_1(m_1-1) + \alpha_2 m_2(m_2-1)}{q} \right)^2 \\
&\leq \left( 1 + d \binom{\alpha_1 m_1(m_1-1) + \alpha_2 m_2(m_2-1)}{d} \right)^2 ,
\end{aligned}
\tag{66}
$$

when $d \leq \frac{1}{2}(\alpha_1 m_1(m_1-1) + \alpha_2 m_2(m_2-1))$. The KL divergence between any two graph pairs in this ensemble is upper bounded by $\epsilon = 12(\alpha_1)\lambda m_1^4 e^{-\lambda(m_1-1)/2} + 12(\alpha_2)\lambda m_2^4 e^{-\lambda(m_2-1)/2}$. We set $d = \lfloor \theta(\alpha_1 m_1(m_1-1) + \alpha_2 m_2(m_2-1)) \rfloor$ for some $\theta \in (0, 1/2)$. Substituting these observations in (2) gives

$$
\begin{aligned}
n &\geq \frac{(\alpha_1 m_1(m_1-1) + 2\alpha_2 m_2(m_2-1))\log 2 - 2(\alpha_1 m_1(m_1-1) + \alpha_2 m_2(m_2-1))h(\theta)}{12\lambda(\alpha_1 m_1^4 e^{-\lambda(m_1-1)/2} + \alpha_2 m_2^4 e^{-\lambda(m_2-1)/2})} \\
&\quad \times (1 - \delta - o(1)),
\end{aligned}
\tag{67}
$$

when $(\alpha_1 m_1(m_1-1) + \alpha_2 m_2(m_2-1)) \to \infty$.

Next, we discuss the values of $\alpha_1$, $\alpha_2$, $m_1$ and $m_2$ such that this ensemble conforms to the class $\mathcal{I}_\eta$. In the class $\mathcal{I}_\eta$, we assume that $\alpha_1 = \alpha_2 = 1$, $m_1 = \lfloor \sqrt{\gamma k/2} \rfloor$ and $m_2 = \lfloor \sqrt{\bar{\gamma} k/2} \rfloor$. Therefore, the total number of vertices used in the shared cluster is given by $2\alpha_1 m_1 = 2\lfloor \sqrt{\gamma k/2} \rfloor \leq \lfloor \eta p \rfloor$ if $\gamma \leq \eta^2 p^2/2k$. Also, the total number of edges in the shared cluster is bounded by $\alpha_1 \binom{2m_1}{2} \leq 2\alpha_1 m_1^2 \leq \gamma k$ and the total number of edges in any graph is bounded by $\alpha_1 \binom{2m_1}{2} + \alpha_2 \binom{2m_2}{2} \leq 2(\alpha_1 m_1^2 + \alpha_2 m^2) \leq k$. Substituting these values of $m$, $\alpha_1$ and $\alpha_2$ in (67), we obtain

$$
\begin{aligned}
n &\geq \frac{((1 - \gamma/2)\log 2 - h(\theta))}{3\lambda k(\gamma^2 e^{-\lambda(\sqrt{\gamma k/2}-1)/2} + \bar{\gamma}^2 e^{-\lambda(\sqrt{\bar{\gamma} k/2}-1)/2})} \\
&\quad \times (1 - \delta - o(1)).
\end{aligned}
\tag{68}
$$