[Reviews · NeurIPS 2019]

Reviewer 1



Major comments: Throughout the paper k is used to refer to the maximum number of edges in a graph, but I'm unclear if that means that the proofs hold for some k = max(k_1,k_2) where k_1 is the number of edges in G_1, etc, and k_1 and k_2 are allowed to be different. If it doesn't allow each graph to have a different value for k this should be made clear. If it does allow that, then it's unclear what ranges of k the proofs hold for (presumably min(k_1,k_2) is lower bounded by \gamma k). Allowing each graph to have a different k implies that there could be different recovery rates for each graph, but the error metric is over the full joint space (rather than the subgraph or independently in the two graphs). Is it possible to make statements about the error metric just in the shared subgraph? As this is the (unknown) region where there is extra information about the graph structure then it would make sense that it's simpler to recover the shared structure, but I don't think any of the supplied results make specific statements about the joint subgraph recovery. Figure 2 is unclear. I believe that the solid line is the ML decoder's sample complexity and the dashed line is the theoretical bound, but the caption and surrounding text could do with clarification. If that's incorrect then I think it could do with more extensive clarification. The experimental evaluations use random graphs with a single specific structure pattern. It would be useful to see more random structures to show how it holds across a wider range of more realistic graphs. Minor comments: The description for figure three says that when the error metric punishes smaller deviations then it's harder to score well on the error metric, which seems tautological. As the error measure changes as d changes this could do with a little more explanation, as otherwise it might lead the reader to think that increasing d to relax the computation allowed the ML decoder to better recover the exact graph (with d = 0). There are several small notational issues, e.g. definition 2 spells out gamma instead of using \gamma in the last line, eq 31 is missing a close paren. The second appendix doesn't use the NeurIPS style file, and isn't folded into the first appendix. Is this intended to be part of the submission?

Reviewer 2



While the paper is interesting and useful, the authors should also compare this paper with recent work at ICASSP that superficially looks similar: https://ieeexplore.ieee.org/document/8682199 I am unaware if the authors happen to be the same, but regardless, the two papers seem very very closely related. If applicable, the authors should update their submission (and their response) with any other material (the above, as well as publications other than the ICASSP paper as well) that may be closely related to their work. I will update my review after author feedback on this point.

Reviewer 3



There are, however, several deficiencies that should be addressed in order for the work to be publishable in a venue like NeurIPS. -- The analysis of the achievability part is based on analyzing (9), which is the ML decoder searched over all pairs of graphs that are \eta-similar. The ML analysis is rather straightforward, involving merely large deviation upper bounds, straightforward counting of graphs, together with those for bounding the KL divergence already present in Santhanam and Wainwright (2012). Hence, it’s not clear what the novelties in the analysis here are. -- The bounds for the converse part are useful but the techniques are based on specializing Fano and using techniques not dissimilar to those in Scarlett and Cevher (2018). -- Even if there are technical novelties in the analyses above, the bounds as shown in Table 1 are not very tight; the sufficient condition(s) seems far from the necessary condition(s). Further, there’s no discussion of this gap. Next, they do not seem to depend on d. -- One thing the reviewer fails to understand is how the authors obtained the plots in Figure 2. Usually such plots are obtained by applying tractable algorithms. However, ML is not tractable even for p = 100. Can ML be implemented in "real-life"? Do these plots show any phase transition in k, p, and \eta? -- Some typesetting is very careless. “gammak” is present in Definition 2. Some compilation Latex errors in the supplementary material. Generally, this submission looks “rushed”. Also, in the supplementary, the reviewer wonders why the fonts for the proof of the achievability and converse parts are different. -- One philosophical comment. Upon reading the title, the reviewer was hoping for recovery guarantees for a *single* graph given side information. However, while what the authors have done here is not incorrect, it is misleading given the title. They recover *both* graphs. Naturally, one would wonder whether one can do better (in terms of fundamental limits) if one seeks to only learn G_1 when samples from G_1 and G_2 are given (and G_1 and G_2 are similar in some sense). I would be thinking this is the true essence of graphical model selection with side information

[Author Response · NeurIPS 2019]

We thank the reviewers for their valuable comments. Relevant points raised are consolidated and addressed together.

**Setting:**

• **Edge constraint (Reviewer 1):** Define $k_i$ as the number of edges in $\mathcal{G}_i$. $k_1$ and $k_2$ can be distinct and take arbitrarily
different values in the range $[0, k]$. We analyze the minimax error rate, i.e., the error performance for the worst-case
combination $(k_1, k_2)$. This represents structure learning in the most *difficult* combination. The minimax rates
take different forms in different parameter regimes, as specified in the paper. Analysis in each regime hinges on
identifying the worst-case combination in that regime.

• **Side Information (Reviewer 3):** We thank reviewer 3 for sharing their perspective on side information, and indeed
their interpretation is also valid. We comment that the case of unilateral side information (e.g,. $\mathcal{G}_2$ serves as the side
information for $\mathcal{G}_1$) is a special case of the bi-lateral scenario that we consider, in which each graph serves as the
side information for the other one. We can recover unilaterial side information results by changing the metric in (7)
from    $\min_{i \in \{1,2\}}\{|E_i \Delta \hat{E}_i|\} \geq d$    to    $|E_1 \Delta \hat{E}_1|\} \geq d$.

**Novelty in analysis (Reviewer 3):** We, respectfully, disagree about lack of novelty in analysis. To furnish a context:
1. Santhanam and Wainwright, 2012 [SW2012] focuses on *exact* recovery and provides both lower and upper bounds.
2. Scarlett and Cevher, 2016 [SC2016] focuses on *approximate* recovery and provides **only** a lower bound.

• **A hitherto un-investigated scenario:** Besides generalizing the regimes [SW2012] of [SC2016] for joint recovery,
we also provide *upper bounds on the approximate recovery*, which is the missing scenario in [SW2012] and [SC2016].
Hence, as a special case of our results we can recover the results for this missing regime for single-graph structure
learning as well. The way different parameter regimes for this scenario are constructed and the ensuing analyses are
distinct from ensemble construction and proofs of both [SW2012] and [SC2016].

• **Generalization of other scenarios** Please note that generalizing the other scenarios from one graph to two graphs
is non-trivial. Even though the ensemble selections are inspired by [SW2012] and [SC2016], their choices and the
techniques for analyzing the minimax rate are different. The similarity in some of the approaches is inevitable (e.g.,
Fano's inequality is pivotal for proving converses in information theory).

• **New insights for ML decoding:** Finally, we note that [SC2016] provides the lower bounds identical or near-identical
to those of exact recovery in [SW2012] for a wide range of edge-bounded Ising models, based on which it was
conjectured that approximate recovery is as hard as exact recovery for the complete edge-bounded subclass. In this
paper, we also establish that this conjecture is true for the edge-bounded subclass for an ML based decoder.

**Relevance to the ICASSP paper (Reviewer 2):** There are significant differences in settings, objectives, and results:
• **Settings** (parameter regimes): The ICASSP paper focuses on very specific Gaussian and Ising models (specific
parameter regimes). In this paper we do *not* consider Gaussian, and focus on a much broader subclass of Ising
models. Specifically, the ICASSP paper focuses on an Ising setting with $k \leq \frac{p}{4}$ and restrictions on the girth and
separation criterion. The results are provided only for the specific regime $\lambda = O(\sqrt{k^{-1}})$ for the relative choices of
$k \leq \frac{p}{4}$ and $\lambda$. In this paper, we consider all values of $k$ and all possible regimes for the relative choices of $k$ and $\lambda$.

• **Objective** (approximate vs. full recovery): The focus of the Ising model section of the ICASSP paper is only on
approximate recovery, while we consider both approximate and full recovery objectives.

• **Results** (necessary & sufficient conditions): The ICASSP paper provides only necessary conditions (lower bounds)
for the specific class mentioned, while we provide both necessary and sufficient conditions for the general settings.

**Sample complexity results:**

• **Shared cluster (Reviewer 1):** Even though not presented in the paper, we can readily show that in *most* parameter
regimes the performance in the shared cluster is similar to single-graph recovery with $p\eta$ nodes and at most $k\gamma$ edges.

• **Results in Table 1 (Reviewer 3):**
– **Tightness:** While we agree that the bounds are not very tight, we also would like to emphasize that that is the
case even for the simpler problems in the literature (e.g., single graph recovery). Our bounds are not any looser
than those for single-graph recovery. Also, in some regimes the results are tighter than others. For instance, when
$k = O(p)$ the difference is only a factor $k$. The mismatch is more profound in denser graphs.
– **Effect of $d$:** The analyses of the necessary conditions in Theorems 2 and 3 and sufficient conditions in Theorem 1
show that $d$ does not affect the asymptotic scaling rate of their respective bounds even when $d$ scales as fast as
linearly with $k$. For instance, in Theorem 1, $d$ appears only in a logarithmic factor scaling at most at the rate of
$\log k$ which is dominated by $k \log p$ in $A_1$ and $A_2$. That is why the summary results in Table 1 do not include $d$.
We will add a comment to highlight this in the final version.

**Numerical results:** Reviewer 1 is correct (the solid curve represent ML in Fig. 2). We will clarify this in the final
version. Also, as Reviewer 1 suggested, we will update Fig. 3 to showcase an average performance over an ensemble
of random graphs. Also, we will provide more explanation on recovery accuracy as we increase $d$ and its interplay
with the computational cost of ML. Specifically, the main observation is that as we *slightly* increase $d$, while the
computational complexity improves *slightly*, the recovery declines *rapidly*. Regarding the question of Reviewer 3,
we note that the numerical evaluations were carried out on a family of sparse graphs, for which the ML estimation
was tractable and implemented via counting the number of instances a vertex has the same value as other vertices. In
the settings we examined, we did not observe any phase transition in the error rate when varying $k$, $p$, $\eta$.

**Typos:** We thank the reviewers for noting the typo on $\gamma_k$, which we will fix. Also, the two appendices submitted as
supplementary documents were generated independently. We will consolidate them for the final submission.

[Meta-Review · NeurIPS 2019]

This paper studies the problem of "simultaneously learning two Ising models whose underlying graphs have some similarity constraints." The problem is interesting (and well-motivated) and the authors provide matching upper and lower bounds, with sharper characterization in some regimes. The proofs use more or less standard approaches, although applying these requires nontrivial work. Overall this is a solid contribution to NeurIPS. The authors responded to most of the reviewer's concerns. This is primarily a theoretical contribution: with a clean problem setting and tight results it has a certain aesthetic appeal which will be appreciated by NeurIPS attendees working on graphical models. The reviewers raised several concerns, many of which were addressed by the authors' response. The authors must address these in the final version of their work. * Make more clear the relationship of these results to the recently published paper at ICASSP: this paper could perhaps be characterized as a (substantive) generalization of that work. * it is unclear what new ideas (that might find use elsewhere) were needed to show the results. While it is true that Fano's inequality is the workhorse for many minimax bounds, their response claims their "choices and the techniques for analyzing the minimax rate are different" without really clarifying what this difference is or whether the choices give some insight into structural aspects of the problem. Since the upper and lower bounds differ, the lower bounds are interesting inasmuch as they give insight into how this specific problem is hard. * There was some confusion about the experimental validation: more careful explanation here would be good. * It is not clear what the practical implications of this result are. In particular, without some understanding of the quality of recovery in the shared subgraph, why is the result interesting outside pure mathematical interest? Tying back to some of the motivating applications would complete the story.